# Post-Fire Impacts of Vegetation Burning on Soil Properties and Water Repellency in a Pine Forest, South Korea

Qiwen Li [1], Sujung Ahn [2,3], Taehyun Kim [1] and Sangjun Im [1,4,*]

1   Department of Agriculture, Forestry and Bioresources, Seoul National University, 1 Gwanak-ro, Gwanak-gu, Seoul 08826, Korea; gimunlee@snu.ac.kr (Q.L.); kta40@snu.ac.kr (T.K.)
2   East Coastal Forest Fire Center of Gangwon Province, 1976 Haean-ro, Gangneung-si 25407, Gangwon-do, Korea; ahnsujung@korea.kr
3   National Institute of Forest Science, 57 Hoegi-ro, Dongdaemun-gu, Seoul 02467, Korea
4   Research Institute of Agriculture and Life Sciences, Seoul National University, 1 Gwanak-ro, Gwanak-gu, Seoul 08826, Korea
*   Correspondence: junie@snu.ac.kr; Tel.: +82-2-880-4759

**Abstract:** Forest fires can have a direct and immediate impact on soil properties, particularly soil water repellency. This study investigated the direct impacts of the Gangneung forest fire of 2019 on soil properties and the spatial variability of soil water repellency with vegetation burn severity in the Korean red pine (*Pinus densiflora* Siebold and Zucc) forest of South Korea. A total of 36 soil samples were collected at depth intervals of 0–5 cm, 10–15 cm, and 20–25 cm from three burned sites, representing surface-fuel consumption (SC), foliage necrosis (FN), and crown-fuel consumption (CC), respectively. An unburned site was also used as a control. Soil properties such as soil texture, pH, bulk density, electrical conductivity (EC), total organic carbon (TOC), and cation exchange capacity (CEC) were analyzed in the laboratory. The increase in the sand fraction near the soil surface after a fire was associated with changes in silt and clay fractions. Moderate to high vegetation burn severity at the FN and CC sites caused a decrease in soil pH due to the thermal destruction of kaolinite mineral structure, but organic matter combustion on the soil surface increased soil pH at the SC site. Forest fires led to increases in total organic carbon at the FN and SC sites, owing to the external input of heat damaged foliage and burnt materials. Molarity of an ethanol droplet (MED) tests were also conducted to measure the presence and intensity of soil water repellency from different locations and soil depths. MED tests showed that vegetation burn severity was important for determining the strength of water repellency, because severely burned sites tended to have stronger water repellency of soil than slightly burned sites. Unburned soils had very hydrophilic characteristics across soil depths, but a considerably thick hydrophobic layer was found in severely burned sites. The soil water repellency tended to be stronger on steep (>30°) slopes than on gentle (<15°) slopes.

**Keywords:** soil water repellency; soil hydrophobicity; vegetation burn severity; topographic gradient; MED test



## 1. Introduction

In the Republic of Korea (hereafter, South Korea), a forest fire is considered one of the most destructive disturbances in the forests. The occurrence of forest fires has increased in recent decades due to heavy fuel accumulation and prolonged warm and dry conditions in winter and spring [1–3]. National fire statistics of South Korea [4] reveal that an average of 437 forest fires have occurred each year, with a damaged area averaging 2050 ha per year during the past 50 years. The causes of these fires were mainly anthropogenic: careless disposal of firebrands (59%), agricultural by-product and garbage burning (18%), and other human activities [4].

Fire alters certain physical and chemical properties of burned soils [5]. Soil heating induces changes in particle size distribution in burned soils by aggregating clay and

silt particles into coarse sand particle [5–7]. Organic matter consumption and loss of macropores (>0.6 mm) also leads to changes in organic matter content, porosity, and bulk density, especially in the top few centimeters of soils [5–7]. Loss of soil organic matter and increased bulk density can decrease the water storage capacity of burned soils. An increase in soil pH is observed after a fire, mainly due to the loss of OH– groups from clay minerals, formation of oxides, or increases in exchangeable cations in soils [5,8,9]. Some studies show that pH decreases in burned soils exposed to high temperature (>500 °C) [5,10]. Fire directly affects the cation exchange capacity (CEC) by the combustion of soil organic matter and destruction of clay minerals [5,11]. The combustion of organic matter at temperature between 100 °C and 500 °C may increase soil electrical conductivity (EC). However, EC is also decreased in soils exposed to temperature of about 500 °C, due to the collapse of the crystalline structure and oxidation [5,10].

A direct effect of fires on the soil environment is the formation of a water-repellent layer on surficial soil or a few centimeters below [5,12]. The combustion of organic matter can release volatile hydrophobic substances, a small part of which moves downward along the soil temperature gradient and coats soil particles, forming a hydrophobic layer when condensed at a cooler part of the soil profile [12]. A thick hydrophobic layer poses significant effects on soil hydrology by hampering water movement through soil layers [13,14], enhancing overland flow [15], and therefore accelerating soil erosion and nutrient loss from fire-affected areas [12,16,17]. By contrast, soil water repellency enhances soil aggregate stability and carbon sequestration [18,19]. The strength of soil water repellency after a fire depends on the intensity of fire. Soil water repellency changes little at soil temperature below 175 °C and increases considerably at temperatures between 175 and 270 °C. However, soil water repellency can be destroyed when soils are heated to temperatures between 270 and 400 °C [12,19]. Though water repellency is produced in all soil textural types, coarse-textured soil, such as sandy soil, is more susceptible to water repellency than fine-textured soil because of the low specific soil surface [20]. McNabb et al. [21] investigated that water repellency gradually decreased with time and returned to nearly pre-fire conditions 6 months after a fire in southwest Oregon.

There are various techniques to measure soil water repellency or soil hydrophobicity. The most commonly used methods to characterize the magnitude of water repellency include the water drop penetration time (WDPT) test [13,20,22,23], contact angle measurement [20,23,24], and the molarity of an ethanol droplet (MED) test [22,25]. The WDPT test is related to the persistence of hydrophobicity, and the remaining two techniques are related to its strength. An in situ measurement of the contact angle between the soil and the droplet is usually difficult under field conditions when the soil surface is not completely flat for contact [24]. The WDPT and MED tests tend to be practically used in the field for measuring the hydrophobicity of post-fire soils owing to their ease of use and lack of need for expensive instruments. The WDPT test measures the time required for a water drop to completely penetrate the soil [22] but is not suitable for extremely water-repellent soils because of the long persistence time for water penetration [22,23]. Compared to the WDPT test, the MED test determines the strength of hydrophobicity, particularly for highly water-repellent soils [25–27], measuring the minimum molarity of an aqueous ethanol droplet that penetrates completely into the soil within a given time (3–5 s) [22,28,29].

The strength of soil water repellency in burned areas has been examined through either field observations [30–32] or laboratory experiments [13,33]. Laboratory experiments have been used to elucidate the relationships between certain influencing factors, such as soil type, fuel type and quantity, soil temperature, soil water content, and soil hydrophobic characteristics under a controlled environment, but they do not always correspond to real fires. Controlled amounts of combustible fuels and the lack of oxygen circulation limit the movement of the hydrophobic substances that induce water repellency in natural environments [34–36]. In addition, most laboratory experiments use sieved or disturbed soils [37–39], which do not always represent field conditions [40]. Conversely, in situ water repellency has been widely measured in various regions, climates, and soil and vegetation

types. However, the occurrence and strength of fire-induced water repellency vary in space and change with time, owing to consecutive rainfall events; therefore, the processes occurring in natural environments are only partly understood due to limited accessibility and time passage after fire events [21].

Terrain features can vary considerably over an area, especially in hilly or mountainous regions. Slopes enhance heat transfer between fire flame and neighboring fuels during a fire [41]. The flame residence time increases with increasing slope angle [42]. Consequently, terrain slope influences fire behavior directly, with the rate of spread being the greatest on steeper slopes [43,44], thereby affecting the water repellency of soils. Fire regimes vary both spatially and temporally with slopes; therefore, the effect of a slope on soil water repellency does not seem to be clearly understood.

The presence and intensity of fire-induced soil water repellency in Europe has been widely reported from Mediterranean forests [15,33,39] and non-Mediterranean forests [30,45–48]. The evidence of fire-induced hydrophobicity was also found in afforested forests of South Africa [32,49], and woodlands [50], mixed forests [44,51], and mixed-evergreen forests [21] of North America, and volcanic ash [28] and peat soils [52] of Japan. However, few studies have been conducted in pine forests of Asian monsoon climate regions.

The objectives of this study were to analyze the variations of soil properties regarding vegetation burn severity and to evaluate the effects of burn severity and topography on soil water repellency. Field measurements on soil water repellency and laboratory analyses of soil properties were conducted, covering a broad range of fire damage and topography in the Okgye forest fire area located on the east coast of South Korea.

## 2. Materials and Methods

### 2.1. Study Area

The study site was located in Okgye Township, Gangneung, Gangwon Province, South Korea (Figure 1a). Gangneung is a city located on the east coast of the Korean Peninsula. The climate is generally temperate and characterized by the East Asian monsoon with humid, hot summers and dry, cold winters. The Korea Meteorological Administration [53] reported an annual precipitation of 1320 mm, with two-thirds of the rainfall occurring during the summer (June–August). The average daily air temperature ranges from −3.2 to 29.4 °C.

The east coast of South Korea, where the Okgye area is located, has distinct characteristics. It has a relatively warm and dry spring season compared to the west, which leads to a very low fuel moisture content. In addition, strong westerly winds that blow over the long mountain range (Baekdudaegan) are also known to be major triggers for large fires on the east coast.

The Okgye area lies in mountainous topography over a broad range of elevations (100–335 m asl) and has a common feature of mountain forest in South Korea. According to US soil taxonomy, the dominating soil type in the area is Inceptisol, which geologically constitutes the Hongjeom series derived from the Upper Carboniferous parent materials [54]. The main vegetation in the area is Korean red pine (*Pinus densiflora* Siebold and Zucc). It is one of the most vulnerable species to fire because of its large amount of volatile resin. Korean pine (*P. koraiensis* Siebold and Zucc) (<10% land area) and oak trees (*Quercus mongolica* Fisch. ex Ledeb., *Q. dentata* Thunb., *Q. variabilis* Blume, etc.) (<5%) are also present. Pine trees have been artificially planted for reforestation since 1980s, while oak trees are naturally generated. The needles and residues of pine trees build flammable fuel beds that cover the forest floors with varying thickness (<10 cm).

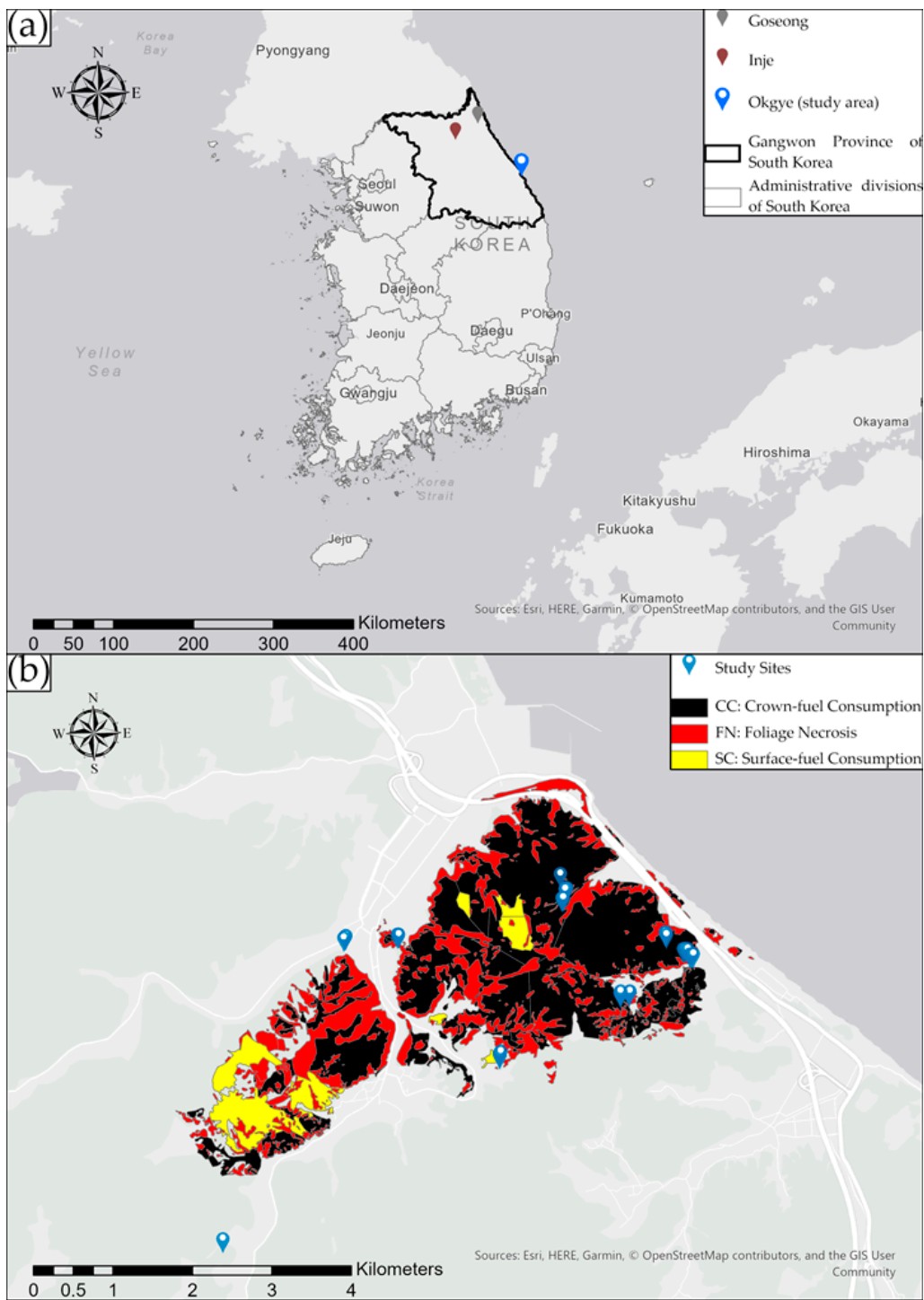

**Figure 1.** (**a**) Locations of Goseong, Inje, Okgye; (**b**) locations of 36 study sites on the vegetation burn severity map (provided by NIFoS, South Korea).

The Okgye area was burned by one of three large fires that occurred simultaneously on 4 April 2019, in Gangwon Province. The three fires occurred in the Gangneung, Goseong, and Inje counties and lasted for three days, burning an estimated area of 2870 ha, including the neighboring areas of Sokcho and Donghae along the east coast. In the Okgye area, more than 1260 ha of forest was severely damaged by the Gangneung fire of 2019 (Figure 2b) [55].

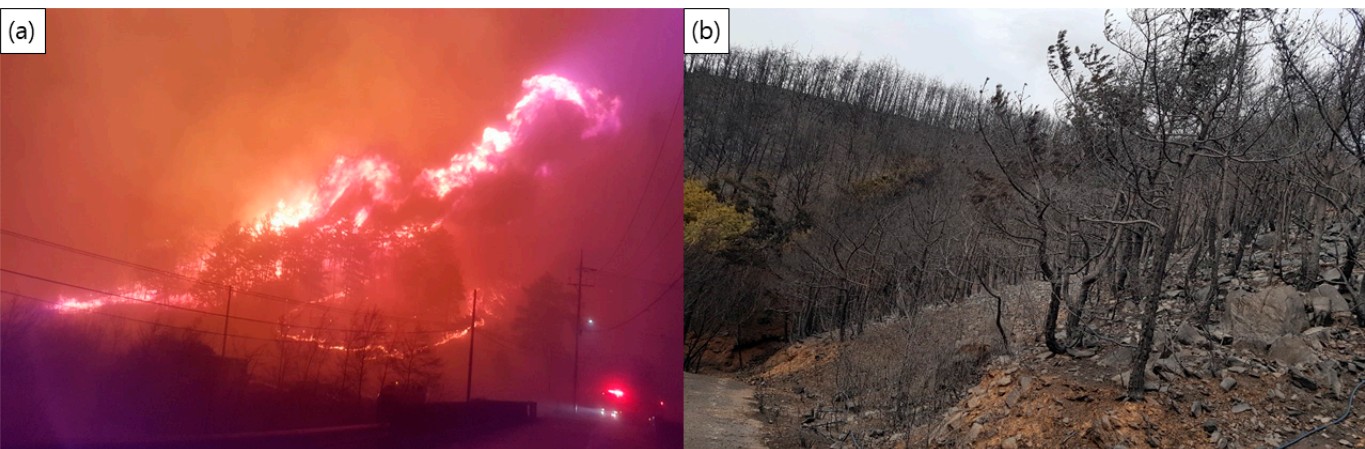

**Figure 2.** (**a**) Front line of the spreading forest fire in Okgye forest on 4 April 2019 (photograph by Gangwon Province), and (**b**) severely burned Okgye forest three days after the fire (photograph by East Coastal Forest Fire Center of Gangwon Province).

In 2019, the prolonged spell of dry weather from winter to spring in the area became a key factor of the large fire. Seasonal precipitation was 57.2 mm in winter (November 2018–January 2019) and 136.6 mm in spring (February–April 2019). This was approximately 39% and 62% lower, respectively, compared to the average precipitation obtained from long-term records. The maximum wind speed and minimum relative humidity on 4 April 2019 were approximately 30 m/s and 16%, respectively, which contributed to the extreme fire spread behavior (Figure 2a).

### 2.2. Field Measurement Design

Approximately two months after the fire event, field measurements were conducted in June 2019. No rainfall was registered between fire event and field measurements, and therefore fire-induced soil properties were assumed to have barely changed. Thirty-six field measurement sites were selected based on the extent of topography and fire damage. Relative humidity and temperature in ambient air during the measurement varied from 24% to 46%, and from 22.5 °C to 30.5 °C, respectively. Wind speeds ranging from 1.5 to 5.2 m/s were observed in the study area at the same period.

The Okgye area has a concave terrain. It is very gentle at the mountain foot and becomes steeper towards the top. Reflecting the change in slope, the burned area was divided into the following three groups in terms of slope gradient: steep (>30°), mild (15–30°), and gentle (<15°).

Fire damage was characterized by the severity of vegetation burn. Vegetation burn severity is a quantitative measure of the effects of fire on the vegetation ecosystem, generally considering the degree of scorching, consumption, and mortality of vegetation, and duff combustion [56,57].

According to the National Institute of Forest Science of Korea (NIFoS) [58], vegetation burn severity in the study area was classified into three classes (Figure 1b), namely surface-fuel consumption (SC), foliage necrosis (FN), and crown-fuel consumption (CC). Table 1 lists the rating descriptions of vegetation burn severity according to fuel consumption and tree crown mortality [58,59]. Here, the unburned (UB) pine forest was also included as a control.

At SC sites, surface fires of low to moderate intensity consumed litter materials, shrubs, and herbaceous vegetation cover on the soil surface. Fires had minor effects on trees at SC sites. At FN sites, low to moderate intensity fires often did not constitute a direct lethal threat to mature trees, but rather may have indirectly caused leaf necrosis by heat-induced injury. At CC sites, high-intensity crown fires consumed live and dead crown fuels, and the combustion of all foliage and small branches in tree crowns caused immediate mortality unless the tree was able to resprout from heat-resistant organs [60,61].

**Table 1.** Vegetation burn severity classification used in the study [58].

| Burn Severity | Fire Intensity | Fuel Consumption and Tree Damage |
|---|---|---|
| Unburned (UB) | No burning | Control, with no evidence of surface fire |
| Surface-fuel consumption (SC) | Low to moderate surface fire | Ground fuel, grass, and shrubs burned, and >60% tree canopy not damaged |
| Foliage necrosis (FN) | Low to moderate crown fire | Canopy partially scorched, and >60% tree crown necrosis due to thermal radiation |
| Crown-fuel consumption (CC) | High intensity crown fire | Canopy completely burned, with ash and charred organic matter deposited on the soil surface |

This study used the vegetation burn severity map that was provided by the NIFoS. A high-resolution satellite image (KOMPSAT-3 imagery taken at 10:14 (KST) on 23 April 2019) was analyzed using the ISODATA method [55]. The results showed that the total burned area in Okgye forest was 1260 ha, of which approximately 45.4% was defined as CC, 14.5% as SC, and 40.1% as FN (Figure 1b). Although the vegetation burn severity may not always adequately reflect the variations in soil heating and fire intensity, the criteria for defining vegetation burn severity are practical for assessing fire damage to vegetation in South Korea.

*2.3. Soil Sampling and Analysis*

In this study, soil samples were collected at different burn severity sites to examine the direct effects on soil properties within a short period after forest fires. Plant litters and ash particles on the soil surface were removed prior to soil sampling. The soil sampler (DIK-1601, Daiki Rika Kogyo Co., Kōnosu, Japan) was used to collect undisturbed soil samples by manually pushing a cylindrical sample can (50.0 mm in inner diameter and 51.0 mm in height) into soil at the surface and at 10 cm and 20 cm depths.

The soil sampling strategy is a critical issue in soil science because of the heterogeneity and complexity of soil environments [62,63]. To obtain the statistical representation cost and time effectively, a stratified random sampling method was adopted in this study. Stratified random sampling involves subdividing the entire site into homogenous subgroups based on the primary investigation. When burn severity patches are accurately grouped into stratified units, sampling in each unit can provide better representation of the spatial variability with a limited number of samples [62,64].

The study area was divided into four stratified units according to vegetation burn severity. For each unit, three sampling sites were selected. As soil properties vary with depth, soil samples were collected at depth intervals of 0–5 cm, 10–15 cm, and 20–25 cm below the soil surface at each site. A total of 36 samples (four burn severity groups, three replicate sites, and three soil depths) were prepared in the study area. All soil samples were transported to the National Instrumentation Center for Environmental Management (NICEM), Seoul National University of South Korea within 24 h for analysis.

Once the samples were carried to the NICEM, they were oven-dried at 105 °C for 24 h and passed through a 2 mm sieve. Soil properties such as soil texture, pH, bulk density, EC, total organic carbon (TOC), and CEC were analyzed in the soil analysis laboratory of NICEM. Soil texture was determined by the micropipette method [65], which is a sedimentation procedure to determine the percentages of sand, silt, and clay content. Soil bulk density was directly estimated from the mass and volume of oven-dried soil samples [66]. Soil pH and EC were measured in a 1:5 soil to water suspension with a pH meter (HM-30R, DKK-TOA, Tokyo, Japan) and an EC meter (CM-25R, DKK-TOA, Tokyo, Japan) [67,68], respectively. The Walkley–Black method [69] and ammonium acetate method [70] were used to measure the TOC content and CEC of the soil samples, respectively.

### 2.4. Soil Water Repellency Measurement

The MED test was used to examine the occurrence and severity of soil water repellency in the study area. Following the classification criteria [22], the degree of water repellency was determined from the volumetric ethanol percentage concentration used in the MED test: class 1, very hydrophilic (0% ethanol); class 2, hydrophilic (3% ethanol); class 3, slightly hydrophobic (5% ethanol); class 4, moderately hydrophobic (8.5% ethanol); class 5, strongly hydrophobic (13% ethanol); class 6, very strongly hydrophobic (24% ethanol) and class 7, extremely hydrophobic (36% ethanol).

Soil water repellency measurements were taken at 36 sites in Okgye and incorporated a range of burn severity classes. Four classes of vegetation burn severity were applied, and nine sites were selected in each class. Three replications were considered for each of the steep, mild, and gentle slopes in each burn severity group. Considering that soil water repellency can vary with soil depth, water repellency was measured at five different depths, that is, at the soil surface and at depths of 1 cm, 2 cm, 3 cm, and 4 cm. In total, 180 measurements were performed in the study area.

### 2.5. Statistical Analysis

Non-parametric methods were applied to test the statistical significance of the results. A Kruskal–Wallis test was used to determine differences in soil properties by burn severity as a nonparametric alternative to the one-way analysis of variance [71]. Since the results from the MED test were categorical data, Fisher's exact test was used to test the differences in soil water repellency among the three burn severity classes and three slope groups with soil depth [72]. The statistical analyses were performed using R (Version 4.1.0, Vienna, Austria).

## 3. Results

### 3.1. Soil Characteristics

Variations of soil textural fractions with burn severity are shown in Figure 3. The soil was characterized by a sandy loam texture according to the USDA textual soil classification [73]. The fraction of sand particles in surface soil layers (0–5 cm) varied from 55.7% to 76.4%, and the fractions of silt and clay ranged from 2.7% to 55.2%, and from 6.6% to 13.8%, respectively. The sand fraction in the surface layer (0–5 cm) was 55.5 ± 8.8% for unburned soils and 63.8 ± 7.1% for burned soils. The fractions of silt and clay particles were larger for unburned soils (30.6 ± 8.6% and 13.9 ± 0.4%, respectively) than for burned soils (27.2 ± 7.5% and 8.9 ± 2.1%, respectively). The fraction of sand particles was relatively similar throughout the depths for unburned soils, whilst it decreased with depth for burned soils (Figure 3). Differences in particle size distribution between burn severity groups at each depth was not statistically significant (Kruskal–Wallis test, $p > 0.05$).

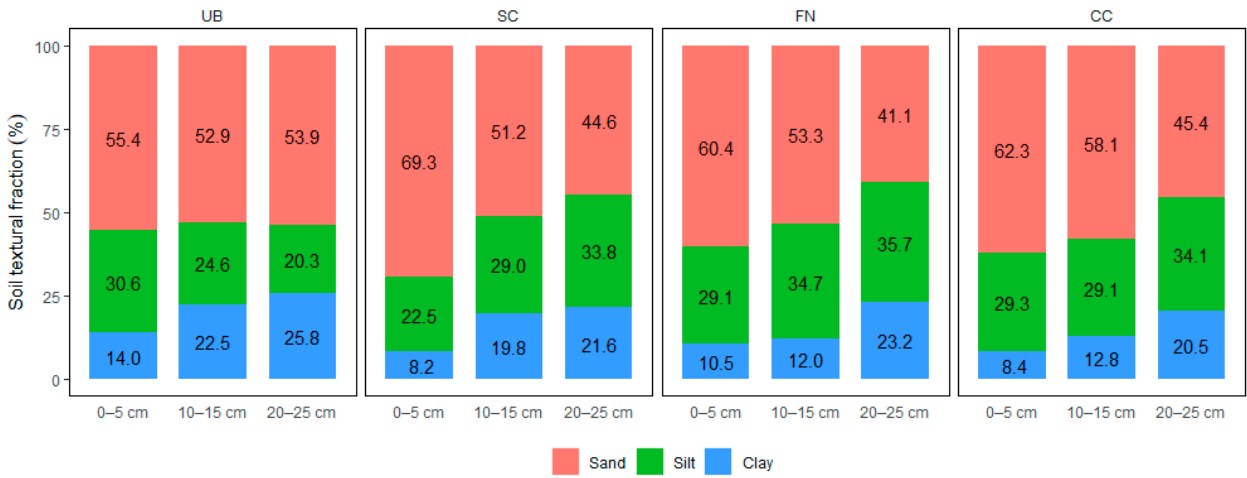

**Figure 3.** Variations of mean (*n* = 3) soil textural fractions (%) with vegetation burn severity (UB, SC, FN, CC).

The soil bulk density varied with depth (Figure 4a). The average bulk density of unburned soils increased with depth from 1.12 g/cm$^3$ at the top layer (0–5 cm) to 1.37 g/cm$^3$ in deeper soils (20–25 cm). This trend did not appear for burned soils, where the highest bulk density was observed at the 10–15 cm depth. For the surface soils (0–5 cm), bulk density of burned soils was $0.97 \pm 0.17$ g/cm$^3$, which was lower than that for unburned soils ($1.12 \pm 0.10$ g/cm$^3$).

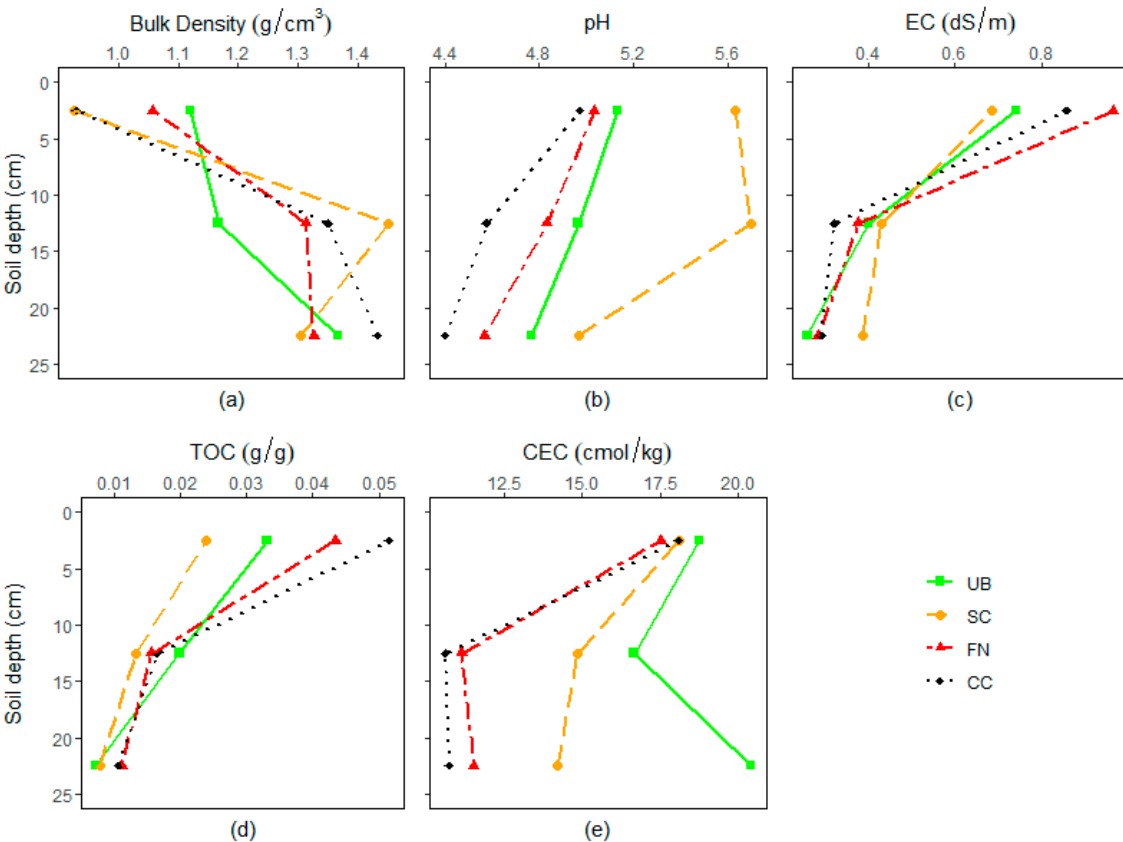

**Figure 4.** Differences in soil properties (bulk density, pH, EC, TOC, CEC) with vegetation burn severity (UB, SC, FN, CC) and soil depth (0–5 cm, 10–15 cm, 20–25 cm).

Soil pH varied but tended to be lower for FN ($4.81 \pm 0.52$) and CC ($4.65 \pm 0.38$) than UB ($4.96 \pm 0.29$) at all depth levels (Figure 4b). The SC, however, had higher pH throughout the soil layers ($5.43 \pm 0.75$) than UB. The magnitude of difference varied with burn severity.

The mean soil EC throughout soil layers for UB ($0.47 \pm 0.25$ dS/m) was lower than SC ($0.50 \pm 0.22$ dS/m), FN ($0.54 \pm 0.33$ dS/m), and CC ($0.49 \pm 0.34$ dS/m) (Figure 4c). For surface soils, SC ($0.68 \pm 0.07$ dS/m) showed lower EC than UB ($0.74 \pm 0.15$ dS/m), while FN ($0.96 \pm 0.17$ dS/m) and CC ($0.86 \pm 0.35$ dS/m) were higher than UB. The magnitude decreased with depth in all burn severities.

TOC was higher at the top 5 cm layers in FN and CC soils compared to unburned soils (Figure 4d). TOC of unburned soils was $0.033 \pm 0.009$ g/g at the surface layer and decreased with depth to $0.020 \pm 0.021$ g/g at the 10–15 cm soil depth and $0.007 \pm 0.004$ g/g at the 20–25 cm depth. TOC of the top 5 cm soil layers varied with burn severity, ranging from $0.024 \pm 0.003$ g/g at SC to $0.043 \pm 0.006$ g/g at FN, and $0.052 \pm 0.020$ g/g at CC sites. TOC tended to be larger for the higher burn severity, with the greatest difference (57.6%) in the soil surface compared to the CC sites.

The burned groups (SC, FN, and CC) showed lower CEC than the unburned group at every depth level (Figure 4e). The CEC tended to be lower at 10–15 cm depth and slightly higher at 20–25 cm depth for all groups. Differences in CEC were more obvious in lower soil layers (10–15 cm and 20–25 cm) than in the surface layer of soils. The CEC values in surface

soils decreased from 18.73 ± 4.73 cmol/kg in unburned soils to 17.91 ± 4.49 cmol/kg, on average, in burned soils after the fire, and decreased from 16.65 ± 11.11 cmol/kg in unburned soils to 10.61 ± 2.02 cmol/kg in CC soils at a depth of 10–15 cm.

### 3.2. Variation of Soil Water Repellency with Burn Severity

Figure 5 presents the vertical distributions of soil water repellency with burn severity. Unburned soils were categorized into very hydrophilic at all depth layers and were therefore excluded from the graphical representation.

At centimeter scales, the differences in vertical distributions of soil water repellency between different burn severity groups were distinguishable (Figure 5). For SC, water repellency was found only between 0 and 3 cm depths, whilst it extended to 5 cm depth for FN and CC. For CC, the top layer (0–1 cm) was mostly hydrophilic (92%), whilst the lower layers showed strong water repellency (MED classes 5–7). The FN group seems to have most vertically spread water repellency throughout the layers amongst the three groups. Extreme soil water repellency (MED class 7) was not observed at SC but was 11% at 1–2 cm depth at FN, and 17% at 2–4 cm depth at CC.

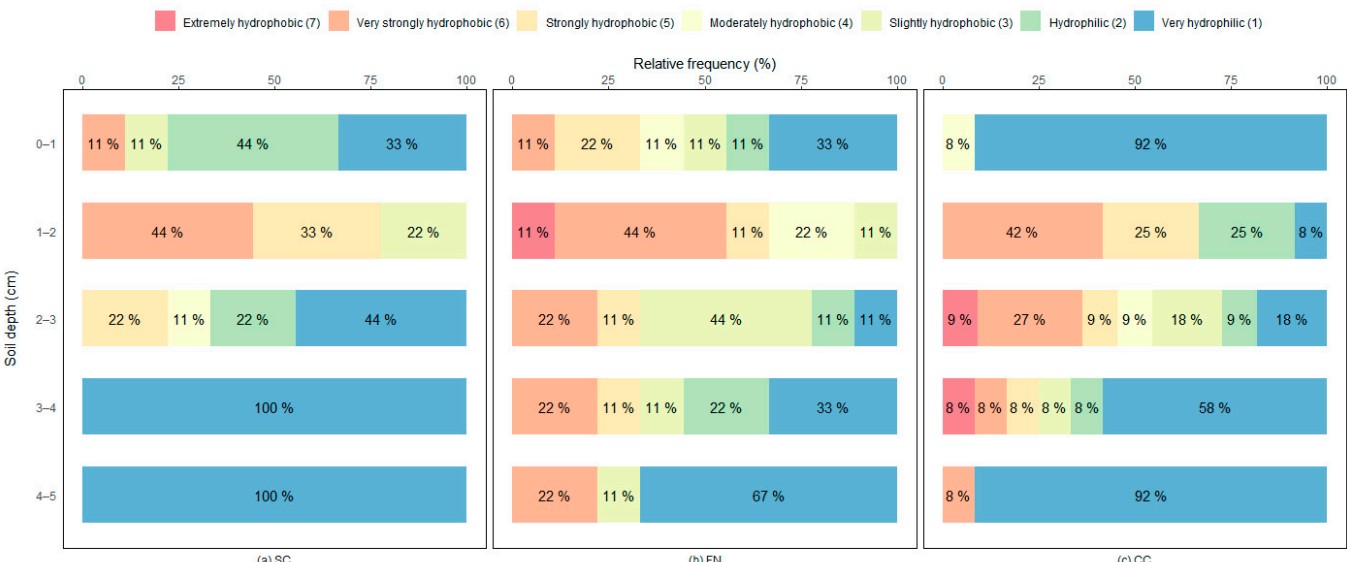

**Figure 5.** Relative frequency (%) of water repellency classes by soil depths (0–5 cm), which were measured by MED tests at different vegetation burn severity (SC, FN, CC) sites. The UB was very hydrophilic for all soil depths and was excluded from the figure.

Table 2 shows the results of Fisher's exact test between the burn severity groups at each depth. For the whole depth of surface layer (0–5 cm), the proportions of hydrophobic soils, which were categorized as 'hydrophobic (MED classes 3–7)', were 31% for SC, 62% for FN, and 37% for CC. This is comparable to the wettable surface of the unburned soils. The FN had a significantly bigger proportion of hydrophobic soils than SC and CC (Fisher's exact tests, *p* values in Table 2).

Table 2 showed that the proportions of hydrophobic soils (MED classes 3–7) at the top surface (0–1 cm) were 22% for SC, 56% for FN, and 8% for CC ($p < 0.1$). At 1–2 cm depth, the proportion of hydrophobic soils for CC was significantly lower than for SC and FN (Fisher's exact test, $p = 0.027$). Approximately 100%, 100%, and 67% of sites for SC, FN, and CC, respectively, were hydrophobic. At 2–3 cm depth, the proportion of hydrophobic soils decreased for SC and FN, whilst it decreased at 3–4 cm for CC.

**Table 2.** Proportion of hydrophobic soils (MED classes 3–7) by soil depths and *p* values of Fisher's exact test on burn severity.

| Layer | Burn Severity | | | *p* |
|---|---|---|---|---|
| | SC | FN | CC | |
| 0–5 cm | 31% | 62% | 37% | 0.007 * |
| 0–1 cm | 22% | 56% | 8% | 0.055 |
| 1–2 cm | 100% | 100% | 67% | 0.027 * |
| 2–3 cm | 33% | 78% | 73% | 0.128 |
| 3–4 cm | 0% | 44% | 33% | 0.076 |
| 4–5 cm | 0% | 33% | 8% | 0.156 |

*: significant at 0.05 level.

### 3.3. Variation of Soil Water Repellency with Topography

Topography also contributed to variations in soil water repellency. Figure 6 depicts the variations in water repellency of burned soils (SC, FN, and CC) on different terrain gradients. When the whole depths (0–5 cm) were treated as one group, soil water repellency was not significantly different between different slope gradients. However, separating the results by each centimeter depth, some differences in soil water repellency were distinguishable (Table 3). At 0–1 cm, the gentle and mild slope groups showed significantly stronger water repellency than the steep slope group ($p < 0.05$). Although no significant difference was found at other depths, the steep slope group seems to have larger proportions of hydrophobic soils at all depths and formation of deeper hydrophobic soil layers than the other two groups (Table 3).

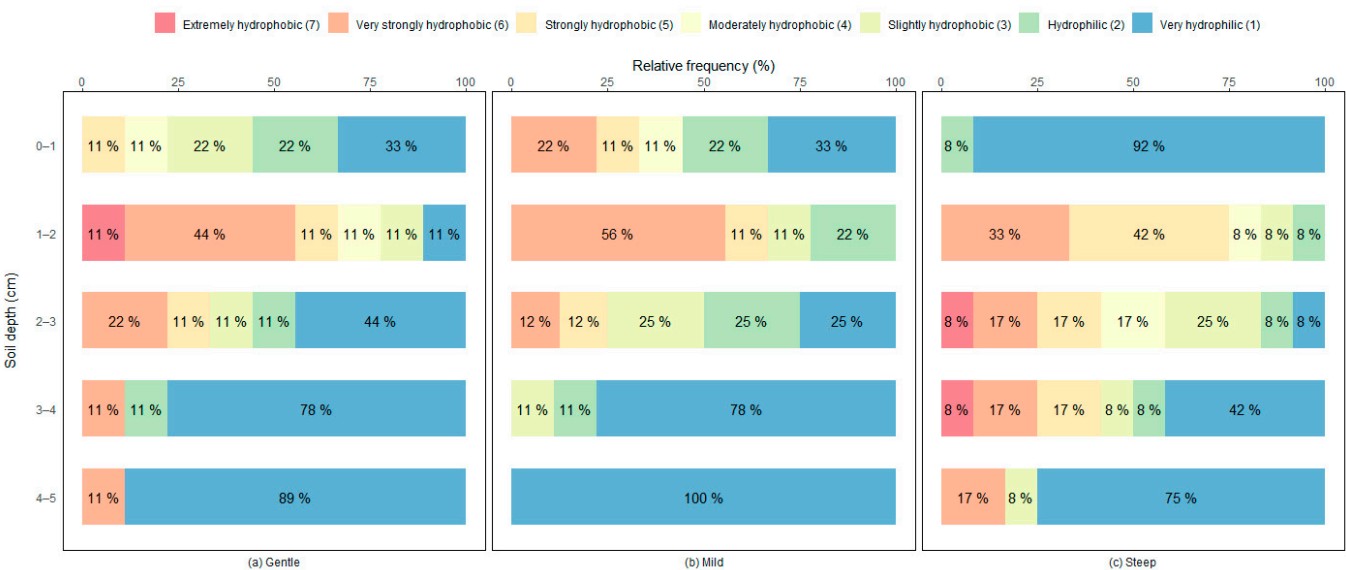

**Figure 6.** Relative frequency (%) of water repellency classes by soil depths, which were measured by MED tests at different slope gradient groups: (**a**) gentle, (**b**) mild, and (**c**) steep slopes.

At centimeter scales, the differences in vertical distributions of soil water repellency between different slope gradient groups were distinguishable. For all slope gradient groups, the largest proportion of hydrophobic soils (MED classes 3–7) was shown at 1–2 cm depth (89%, 78%, and 92% for gentle, mild, and steep slope groups, respectively). For the steep slope group, the top layer (0–1 cm) was hydrophilic (100%), whilst the lower layers showed a large proportion of strong to extreme hydrophobicity (MED classes 5–7). The mild slope group seemed to have the largest proportion of hydrophobic soils on the top surface (0–1 cm) amongst the three groups, but no water repellency was found at 4–5 cm depth.

Table 3. Proportion of hydrophobic class of MED test by soil depths and the results of Fisher's exact test on slope gradient.

| Layer | Slope | | | p |
|---|---|---|---|---|
| | Gentle | Mild | Steep | |
| 0–5 cm | 40% | 38% | 49% | 0.464 |
| 0–1 cm | 44% | 44% | 0% | 0.015 * |
| 1–2 cm | 89% | 78% | 92% | 0.805 |
| 2–3 cm | 44% | 50% | 83% | 0.169 |
| 3–4 cm | 11% | 11% | 50% | 0.137 |
| 4–5 cm | 11% | 0% | 25% | 0.348 |

*: significant at 0.05 level.

For the whole depth of the surface layer (0–5 cm), the proportions of hydrophobic soils (MED classes 3–7) were 40% for the gentle slope group, 38% for the mild, and 49% for the steep (Table 3). At the top surface (0–1 cm), the proportions of hydrophobic soils (MED classes 3–7) were 44% for both the gentle and the mild groups, and 0% for the steep slope group, which were significantly different at the 0.05 level (Fisher's exact test, $p = 0.015$). The proportions of hydrophobic soils were highest at 1–2 cm depth for all slope groups.

## 4. Discussion

### 4.1. The Impacts of Vegetation Burning on Soil Properties

Fires can substantially alter soil physical and chemical properties to various soil depths depending on the severity of the fire [5]. Burned soils had larger fractions of sand particles than unburned soils at the surface (0–5 cm), which decreased with depth. Variations of the sand fraction among the four sites with different burn severities seemed to be associated with changes in the silt and clay content. This result agreed with previous studies in which fire-induced aggregation of fine particles resulted in increased fractions of coarse sand-sized particles [5–7,10]. Another explanation might be associated with enhanced selective loss by surface erosion or eluviation of finer particles due to reduction of soil aggregate stability by combustion at high temperatures [74].

Soil bulk density nearly increases with a corresponding decrease in soil porosity after a fire [11,75], but this study seemed to be contrary to previous studies. This was because of finer (<2 mm) ash particles that may penetrate and mix with soil materials after a fire. Thus, soil bulk density in the soil surface of burned soils was lower than unburned soils, and it may have increased with depth as a result of soil compaction.

The extraction of Al and Si oxides from the kaolinite, which is the major clay mineral in the study area [54], during fire contributed to decreases in soil pH at both FN and CC sites. This proves the conclusion made by previous studies [5,8,10], which reported that soil pH decreased when clay was exposed to high temperatures. The thermal alternation of kaolinite mineral structures occurs at temperatures over 600 °C [8]. Complete combustion of organic materials on the forest floor at the SC site could increase soil pH by binding H+ in the soil surface [5,76,77].

Soil EC was higher at FN and CC sites compared to the unburned site as a result of the release of inorganic ions from the combusted organic matter (ash) [5,8,77]. By contrast, the formation of coarse particles can decrease soil EC [8,10]. A higher sand fraction (69.3%) in the surface (Figure 3) induced lower soil EC at the SC site than FN and CC sites. Fires can induce changes in the TOC of burned soils, but their impacts are highly variable, depending on fire severity. Combustion of surface fuel caused the reduction or total removal of organic matter on the forest floor and thereby the decrease in TOC at the SC site. Substantial consumption of organic matter begins at temperatures between 200–250 °C and is complete at temperatures around 460 °C [78]. At the FN site, an increase of TOC in the soil surface was observed due to a substantial incorporation of canopy necrosis. Rashid [79] obtained similar results in a Mediterranean oak forest in Algeria. An increase in TOC at the CC site is assumed to be related to the increased deposition of charred materials in the soil surface

as a consequence of external inputs, mainly burnt materials in the tree canopy by crown fires [80]. The changes in soil TOC occur in the upper 10 cm soil layer [80] because organic matter concentrates on the soil surface [5,8].

Fire seems to have decreased the CEC of soil at every depth level. Fire can directly affect CEC by the combustion of soil organic matter and the transformation of clay minerals [11]. The combined effects of organic matter consumption and thermal structural decomposition of clay minerals may induce non-significant differences in CEC at the soil surface. At the 10–15 cm soil depth, the lowest value of CEC was observed at the CC site, followed by the FN site, due to the burning of organic matter. The thermal destruction of soil organic matter contributes to the decrease in soil CEC. The CEC will be easily recovered with vegetation succession on burned areas, but the loss of CEC due to mineral alteration is longer lasting, though limited in spatial extent [75,81].

Fires may cause several changes in physical and chemical properties of soils. The magnitude of these changes is highly related to the combustion of organic matter and thermal modification of soil aggregates, which are controlled by fire severity, fuel type and quantity, and soil characteristics. The external input, such as heat damaged foliage at the FN site and partly or charred plant materials in the tree canopy at the CC site, can affect the organic matter on the soil surface. The results obtained confirm the previous findings, but the effects of vegetation burn severity on soil properties were not straightforward, due to a small sample size considering the inherent spatial variability of soils.

### 4.2. The Impacts of Vegetation Burning and Topography on Soil Water Repellency

Korean red pine is the dominant tree species in the study area and contains a large amount of volatile pine resin. Needle and leaf combustion releases a greater amount of volatile organic compounds than branches and litter combustion [82]. Unlike deciduous trees, pine trees have needles attached to branches in early spring, and this contributes to the severe water repellency of burned soils in pine forests. The differences in the severity of water repellency between tree species can be insignificant in field or laboratory measurements of soil water repellency [30,83,84]; however, the volatile hydrophobic compounds, including resin, wax, and aromatic oils, can induce more severe water repellency [85].

Severe water repellency of soils was found in the study area. It seemed to be affected by the soil texture, which was primarily sandy loam. The soil grain size distribution affects water repellency [20]. Coarse sand is highly susceptible to water repellency because it takes less hydrophobic material to coat the small surface area of the particles compared to silt or clay textures [86]. The severity of soil water repellency increases with a decrease in clay content, resulting in significantly extreme water repellency in sandy soils [20,87].

For burned sites, strong soil water repellency was found most frequently near the soil surface, and less frequently with soil depth, which seemed to be due to limited heat transfer through soil [31,88,89]. Due to the low thermal conductivity of soils, it is expected that the direct and immediate effects of a fire are apparent only at the first few centimeters of the soil column [19,90], where most hydrophobic substances condense [19,88,91].

Vegetation burn severity influenced the strength of soil water repellency and the thickness of repellent soil layers. Surface fires (SC) induced less severe water repellency and thinner layer of hydrophobic soils on the surface soil (0–5 cm) than high-intensity crown fires (CC) or the medium-intensity crown fires (FN). It seems to be due to the lower intensity of heat and shorter flame residence duration for surface fires than crown fires. These results concur with those of many previous studies [20,46,49,92,93], demonstrating that the strength of water repellency depends on the soil temperatures reached during burning. Regardless of fire severity and soil feature, they also reveal that the thickness of the water repellent layer rarely extends 6–8 cm below the soil surface [12,88]. The thickness of water repellency measured in burned soils was within the range of previous results.

The longest duration of the flame residence is assumed for crown fire sites, where forest fires completely burn a thick, combustible litter layer and aboveground vegetation. Previous studies showed that temperatures in crown fires reach over 850 °C above the soil

surface [12] and are observed to be 175–180 °C at the 2–5 cm soil depth [94]. The heat fluxes that reach the soils seem to have produced relatively strong water repellency in the deeper soil layer (2–4 cm) at those sites. It is explained that higher soil temperatures, between 175 and 200 °C, can lead to the translocation of hydrophobic substances downward along temperature gradients and produce water repellency in deeper soil layers [12,92].

Topographically, the slope was a minor predictor of burn severity and the associated soil water repellency. Regardless of vegetation burn severity, topographic steepness increased the water repellency of soils, with steep terrains exhibiting higher water repellency than mild slopes. The effects of topographic gradient on water repellency were more obvious in soils on steep slopes than on gentle and mild slopes. On steep slopes, fire spreads faster [43], and with combined radiation and convection, heat can lead to stronger water repellency in soil layers. With the slope angle, the advancing flames are tilted closer to the surface upslope, which can increase convective and radiant heating on to the surface fuel [42]. Increased flame length and intensified surface fire can also lead to higher temperatures or longer residence time on the soil surface. All these factors contribute to elimination of water repellency at the soil surface and formation of a hydrophobic soil layer underneath the surface [12,92].

## 5. Conclusions

The forest fire in the pine forest (dominated by *P. densiflora*) of South Korea posed physical and chemical impacts on soil properties. Soil heating can cause the thermal alteration of soil aggregation during a fire as a result of the collapse of the kaolinite structure. The burned sites in the study area tended to exhibit stronger soil water repellency than the unburned sites. The effect of topography on water repellency was distinguishable, though it was not statistically significant, implying that steeper slopes tended to have severer soil water repellency. The negative impacts of organic matter consumption can be easily recovered through vegetation succession on burned forests, but the effects of clay mineral disruption are longer-lasting. Therefore, it is necessary to know what are the changes that fires cause to physical and biochemical properties of soils, the causes and consequences of soil property alterations in forest ecosystems, and the recovery processes that occur on burned soil environments.

**Author Contributions:** Conceptualization, Q.L. and S.I.; methodology, S.A.; software, Q.L.; validation, Q.L., S.A. and S.I.; formal analysis, Q.L.; investigation, Q.L. and T.K.; resources, S.A.; data curation, Q.L.; writing—original draft preparation, S.I.; writing—review and editing, Q.L., S.A., and S.I.; visualization, Q.L.; supervision, S.I.; project administration, S.I.; funding acquisition, S.I. All authors have read and agreed to the published version of the manuscript.

**Funding:** This work was supported by a National Research Foundation of Korea (NRF) grant funded by the Korea government (MSIT) (2019R1A2C1089203).

**Acknowledgments:** We would like to thank Choong-Shik Woo and Keunchang Jang from the NIFoS for providing the vegetation burn severity map.

**Conflicts of Interest:** The authors declare no conflict of interest.

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
