# Peer review of "Post-Fire Impacts of Vegetation Burning on Soil Properties and Water Repellency in a Pine Forest, South Korea"

_forests, doi:10.3390/f12060708_

Round 1

Reviewer 1 Report

Manuscript forests-1181037 investigated the direct impacts of a forest fire on soil properties and the spatial variability of soil water repellency with vegetation burn severity in a pine forest in South Korea. Findings are not extremely novel, but they do fill a regional knowledge gap. The introduction is well-written and concise. The methods are exceptionally clear compared to most manuscripts I've reviewed on the subject. Results are well constrained and the author does not over-extend the discussion or conclusions. I do not often compliment manuscripts in review, but this is one of the most straightforward manuscripts I have read in a very long time and that was refreshing. Before the paper is accepted, there are few questions authors need to consider.

  1. Authors need to consider if the title of this paper can be revised to: The Post-fire Impacts of Vegetation Burning on Topsoil Properties in a Pine Forest, South Korea?

2. The introduction section should be enriched, e.g. in line 65-66, authors should give specific examples of the previous studies, which conducted in Europe, America and Africa, and make a comparison of these researches, differences and similarities.

  1. In MM section, authors should provide the information of Relative humidity in the study area, for this enviromental factor has a great impact on the forest fire spreading.
  2. Line 121-122, these sentences should be deleted.

Reviewer 2 Report

A careful read for grammar and sentence structure is needed.

Abstract

"An increase in the sand fraction was observed in the surface soils at burned sites and was associated with decreases in the silt and clay fractions." (lines 16-17) is not supported in the research. If there were pre-fire and post-fire tests done then this is a possible statement. Perhaps the authors were attempting explain differences among the four classifications of fire condition sites, which could be described in another sentence.

Methods

Study area 

Are these urban forests, or forests on the urban/wildland interface or rural forests? The description indicates these are in the city of Gangneung. If these are urban forests, then there is body of literature that needs to be reviewed in the introduction.

Soil Sampling and Analysis

Drop first sentence of second paragraph (Lines 152-153), then combine first and second paragraph.

Drop opening text of third paragraph "In this study," (line 152)

At end of fourth paragraph, add references to specify which methods were used.

Statistical analysis

Add references to each sentence of paragraph.

Drop text "(ANOVA)" in line 200 - the abbreviation is  not used elsewhere in the text.

Results

Line 255 drop the word "was".

Line 278 drop the word "were".

Line 302 add the word "the" between the words "of surface".

Discussion

Drop the sentence " Increase of the sand fraction after the fire seems to 319
be associated with decreases in the silt and clay content." in lines 319-320. Again no pre-fire measurements were made. To correct the related sentence in the Abstract, I suggest reconsidering the meaning of the paragraph and rewriting the abstract sentence.

The final paragraph of the discussion is tangentially related to the other parts of the discussion and is not supported by references and stands separate from the remainder of the discussion. I admire the point of the closing paragraph; however, the point is nearly totally unconnected to manuscript - not included in the literature review in the introduction nor related to the research in the methods or results. 

Conclusions 

The closing paragraph of the discussion has the same problems as the closing paragraph of the discussion (see concerns in discussion comment) but stands out far more because it is a third of the conclusions.

I recommend expansion of the text of all sections of the paper to integrate these important points. But failing this expansion drop the final paragraph of the Discussion and the final paragraph of the Conclusions, which would be sad losses.

Reviewer 3 Report

A report for forests-1181037, entitled “Post-fire Impacts of Vegetation Burning on Soil Properties in a Pine Forest, South Korea”

****General comments****

This paper presents valuable information about the effect of a forest fire on (physical and chemical) soil properties in a Pinus densiflora dominated forest in eastern South Korea. The authors selected four treatment groups (control/unburned UB, SC, FN and CC) based on the burn severity/fire intensity and collected soil samples two months after the fire event in 2019. The authors focused on soil properties (pH, EC, CEC, TOC, bulk density) and soil hydrophobicity. They tried to explain the differences along a depth profile (0-25 cm and 0-5 cm, respectively) between the treatment groups and topographic gradients (gentle, middle, steep) using (one-way) ANOVA.

The authors found out that fire causes a soil hydrophobicity in the upper few cm of the fire-affected soil which they associated with the vegetation type and the soil texture. The higher the burn severity, the thicker the water-repellent layer. The authors could not find a clear (significant) relation between topography and formation and thickness of a water-repellent layer, but assume that steeper slopes lead to a more pronounced soil hydrophobicity.

The manuscript is written well and the figures are a plus for this manuscript, but it lacks a certain “depth” and a lot of information (especially in Introduction and Materials and Methods). Most information is only discussed superficially. Results about the EC are neither mentioned nor discussed. The authors present no hypotheses or objectives. The novelty needs to be more pronounced. The methods (soil sampling, sample treatment in lab) are not mentioned. The Conclusions are mainly a summary, but don’t really synthesize the discussed results or give a perspective. The Discussion is one text, but should be divided into subsections to guide the reader. Several aspects are missing to complete the picture and to get a connection to the “forest ecosystem”: such as the relevance of the forest ecosystem in such a monsoon-influenced area with dry and wet seasons, how the climate change would be relevant, chances of a soil recovery, influences on the function of this ecosystem, consequences on the ecosystem’s resilience or stability, possible short-/long-/permanent-term changes in the forest soil, consequences for other physico-biogeochemical variables.

It is not clear to me, why the authors haven’t measured the soil temperature and moisture before the hydrophobicity test. This might have explained the variations.

Certain terms and information are not mentioned, which would be helpful to introduce and discuss this matter. Examples are: post-fire only occurs in the title but not in the text; the term wildfire; fire effects in forest soils are always defined not only by burn severity and fire intensity but also by soil type, forest type, season, duration, weather and climate, frequency of fire, etc. Why is this not mentioned in the introduction? I am also missing information on e.g. the temperature range when hydrophobic layers are formed and when they are destroyed, the temperature range when soil particles such as sand and silt are changed or destroyed (1414°C for sand, 460-980°C for clay). There are some publications stating that the increase in sand and decrease in silt content in soils is connected to soil temperatures >600°C (see for example review by Ngole-Jeme 2019). Please use such (and other) information to build up the discussion part.

This is a very interesting and important research topic, but the manuscript needs much more and thorough work in order to be ready for publication.

****Specific comments****

Title

-The term “post-fire” occurs in the title, but is hardly mentioned in the text.

-I think it would be helpful to mention the term “water-repellency” or “soil hydrophobicity” in the title. The title itself is very general and vague and therefore, may be not of interest for readers who are looking for publications about “soil hydrophobicity”. The “soil hydrophobicity” is supposed to define the novelty of this manuscript, isn’t it?!

Abstract

-Please use a uniform style by using only one or two terms for the fire event. The authors use too many terms: “vegetation burning” (title), “forest fire” (L14), “surface fire” (L383), “crown fire” (L136).

-The authors fail to mention the materials and methods (area in South Korea, Vegetation type, analyzed soil variables, number of soil samples, used methods ).

-It is suggested to end the abstract with a perspective/ conclusion.

L22 – That is just an assumption and was not measured or analyzed by the authors.

According to the Forests guidelines, the authors could use a max. of 300 words for the abstract. The authors only used 166 words. Hence, I strongly recommend to use up to 300 words and add more information to the abstract.

Keywords

---

1. Introduction

L28-35 – I suggest to begin the introduction with the topic fire, its effects, etc. and then add information (from this paragraph) about the location and forest type used in this study.

L28-29 – The authors mention that fires in the Republic of Korea are connected to the forest characteristics, climate and topography. How? Could the authors please add more information.

L36 – What do the authors mean by “principle”?

L36 – Please be more accurate about the reasons for fire. The term “environmental problems” is very vague and needs more preciseness and/or more facts.

L37-39 – I don’t agree. A water-repellent layer can be formed during a fire, but this is not always the case. The formation of a water-repellent layer strongly depends on the soil temperature. In addition, such a water-repellent layer can also be destroyed by high soil temperatures. Hence, please be more precise about this statement.

L47 – What are those “certain influencing factors”?

L71 – It is important to mention that the Okgye forest is located on the eastern coast of the Republic of Korea (South Korea).

Since this is a study dealing with soil hydrophobicity, I strongly suggest to add more information about the (known) conditions, processes and consequences to the soil and the forest ecosystem. The information presented in this manuscript is very little and very vague.

Why don’t the authors mention the influencing factors such as soil type, forest type, season, duration of fire, etc.?

The authors just focus on soil hydrophobicity in the introduction and do not mention (state-of-the-art information of) soil characteristics such as pH or bulk density. Why not?

The authors do not mention what methods the will use.

Where are the objectives or hypotheses?

2. Materials and Methods

2.1 Study Area

L78 – I suggest to round the number to 1,320 mm.

L81 – The authors mentioned in the first sentence of the introduction that they will use “South Korea”. Hence, please do so instead of writing “Korea”.

L87-90 – Please add the botanic author citation to each species (e.g. Pinus densiflora SIEBOLD & ZUCC.).

L89-90 – If a botanic name such as “Pinus” and “Quercus” is mentioned once, it can be abbreviated thereafter. Hence, the authors should write “P. koraiensis BOTANIC AUTHOR” and “Q. dentate BOTANIC AUTHOR, Q. variabilis BOTANIC AUTHOR”. 

L91 – Please write “varying thickness” instead of “varied thicknesses”.

L97 – Please write “a high-intensity fire” instead of “high intensity of fire”.

L97 – The authors need to define the term “high intensity”.

L98 – The authors could write the letters “(a)” and “(b)” at the inner top of the respective photographs and delete this line since the authors explain the photographs in the figure caption.

L99 – I suggest rewriting the description for Fig. 1a. Suggestion: “Front line of the spreading forest fire in Okgye forest on April 4, 2019”.

L100 – Please add the date or time span (e.g., 2 days after the fire) to the figure caption.

L102 – What do the authors mean by “prolonged spell”? Please explain or adjust.

Figure 2 should be mentioned in the very first paragraph of Materials and Methods (and should be renamed Figure 1). Maybe add the location of Goseong, Gangneung, and Inje counties.

Is there any information on the tree age and understory vegetation? How is the forest used?

Any information on the soil type? What is the underlying bedrock?

How was the forest fire (on April 4, 2019) initiated? Lightning, human activities, or unknown reasons? How many forest fires are caused naturally in the Republic of Korea and how many are caused by human activities (camping, smoking, etc.)?

2.2 Field Measurement Design

L117 + Table 1 – The authors need to add the references to the caption or table. Where do they base the definition on?

Figure 2 – Please add the north arrow and scale to the map of the Republic of Korea. Add a legend that the red dot represents the location of study area and that it is shown more detailed in the satellite/aerial images.

Please add a legend for the colors in Fig. 2a (red, green, etc.).

Please highlight the blue dots in Fig. 2a (make them more pronounced because they are difficult to see). I suggest using colors that could be seen by color-blind people (e.g., red and green could both appear brown to a color-blind person).

L130-131 – Delete the extra captions under every image because the information is mentioned in the actual figure caption. Please add the term “South Korea” to the caption. Use “South Korea” instead of “Korea” (after “NIFoS”).

L132-137 – Please use the past tense since the consumption of the fuel has already happened.

L140 – Please mention the figure in the first sentences instead of in L143.

L142 – I suggest writing “forest” instead of “forestland”.

2.3 Soil Sampling and Analysis

A lot of information is missing. How were the samples taken? With a corer? Add information on the type, manufacturer, location as well as the dimensions (diameter, length).

Why only the soil depth 0-5, 10-15 and 20-25 cm? What about 5-10 and 15-20 cm? The interpretation of the data may not be complete because 2x 5 cm of the soil column is missing.

What happened to the soil samples in the lab? Were they dried and/ or sieved? Which drying temperature? Which mesh size?

How did the authors measure the bulk density, soil texture, CEC? How did they analyze the soil pH, TOC content and EC? Which methods and equipment did they use?

There is no information of the ash layer? Which color? Thickness? Did the authors brush the ash layer off to take samples from the mineral soil? Did the authors found BC?

2.4 Soil Hydrophobicity Measurement

L170-187 – This is information that should be mentioned in the introduction, but not in the Materials and Methods section. The authors present several methods and add pro and cons to every method. Please avoid that in this chapter. Just mention the used method and explain it or highlight the advantages. The authors used only the MED test, correct?!

How did the authors separate a soil into 1cm-sections?

L195 – What is “viz.”?

2.5 Statistical Analysis

Which statistical program was used? Which version and which packages?

Why didn’t the authors choose to correlate some results and gain information on a possible relationship between certain (chemical and physical) soil properties?

3. Results

3.1 Soil Characteristics

L205 – This is the results sections, not the discussion. Hence, please delete this sentence completely. The authors present the data in the results, but discuss and interpret them in the discussion part.

L206 – The soil can either be “characterized as” a specific soil type or be “characterized by” a certain texture.

L216 – Please be more accurate. It should be “Figure 4b” instead of “Figure 4”. Why do the authors begin with the bulk density which is 4b? Either change the sequence of the 5 plots in Figure 4 or begin with the soil pH (4a).

L217 – Please add a space between the parenthesis and “to”.

L218-219 – This is an interpretation and does not belong to the results section.

Why didn’t the authors define the soil type?

The numbers can be rounded to one digit after the comma.

Are the presented numbers mean values?

Please check the numbers again. E.g., if the numbers of the figure are used, the sand fraction of the burned soils is 53.97% and not 63.83%.

Figure 3 – Please add a space between the variable names (Soil texture, Sand, Silt, Clay) and the unit %. Why is the unit mentioned after the soil fractions, since it is already mentioned at the y-axis label?

L224 – The abbreviations UB, SC, FN, and CC need to be mentioned in the caption as well as the unit and the soil fractions. Please be more accurate.

L225 – Figure 4a needs to be mentioned.

Figure 4 - Please add a space between the variable names (EC, TOC, CEC, Bulk Density) and the respective unit. Please use different styles of lines to make them more distinguishable. Please put the legend e.g. to the right of the plot (e).

Why is TOC shown in % and is not related to the area (e.g., g m-2)?

The unit of the bulk density is wrong (the authors wrote "cm3").

L229 – Please add more information (which soil properties, which burn severity level, units) to the caption.

L230 – Figure 4c needs to be mentioned.

L231-232 – This is an interpretation and does not belong to the results section.

L238 – Figure 4 d needs to be mentioned.

L244-246 – This is an interpretation and does not belong to the results section.

What about the mentioning of the EC?

3.2 Variation of Soil Hydrophobicity with Burn Severity

L247 – Why using “Soil Hydrophobicity” here, but “Soil Water Repellency” for subchapter 3.3? Please stick to one term because the data is based on the same method (MED).

L251-260 – Some information is mentioned twice in this paragraph. The order of information is very confusing and does not follow a clear pattern. Please adjust.

Figure 5 - Please add a space between the variable names (Relative frequency, Soil depth) and the respective unit.

L262-263 – Please add the information from L249-250  to the caption (e.g., "UB was excluded as it was defined as 100% hydrophilic throughout the soil column 0-5 cm."). Add more information (which repellency classes, which soil depth, which sites).

L264 – The Fisher’s exact test should be mentioned in the Materials and Methods.

What about the initial water content of the soil? This might had an impact on the soil hydrophobicity.

What about the weather on the day of testing the soil hydrophobicity? Wind, humidity, solar radiation?

Table 3 – I suggest visually separating the 0-5 cm results from the single soil layers by e.g. a space row or a dash line.

L274 – I suggest writing “(p < 0.1)” instead of “(significant at 0.1 level)”.

L278 – I suggest placing the reference to Table 3 in L273-274.

3.3 Variation of Soil Water Repellency with Topography

L279 – Please see comment for L247.

L286 – Please see comment on L274.

L294 – Please add “Table 4”.

L296 – Please add a space between “2” and “cm”.

Table 4 – See comment on Table 3.

4. Discussion

Why didn’t the authors use subsections in the Discussion part? If the authors have objectives or hypotheses they could use them to guide the reader through the discussion. The discussion presented by the authors does not follow a clear pattern.

The authors fail to connect all results to tell a story. What about soil temperature ranges that may have led to the destruction of the silt fraction?

The authors present data about (burned and unburned) forest soil, but hardly connect the data to the fire-affected forest ecosystem.

L311 – What do the authors mean? Physical, chemical or biological soil properties? Please be more accurate.

L313-317 – This section is not very flattering for the presented study. The authors should not begin their discussion with downgrading their results by stating that the quantity is awful. It gives the impression that these results are not worth being published. The authors can discuss this aspect later in the Discussion section.

L324-325 – Please rewrite this sentence.

L326-327 – This is the first time “ash” or “black carbon” is mentioned. Please add information to the Materials and Methods if the authors found ash layers or analyzed any BC. If the authors have not measured/analyzed the ash layer or BC, they should rewrite the sentence because they can only assume that the increase in TOC is connected to ash or BC.

L330 – Please discuss that the pH does not always decrease after a fire and mentioned contradicting studies (pH can also increase after a fire due to the release of certain elements).

L336 – Please write “have” instead of “has”.

L347 – It was not mentioned before that the soil texture is defined as a sandy loam. Please do so.

L352-356 – Please add more information, discuss more and be more specific.

L366 – I disagree. A crown fire can also occur when the litter layer is not thick.

L373-382 – What about the daily temperature range and differences between the three slope areas (gentle, medium steep)? This may influence the soil hydrophobicity two months after the fire.

L386-387 – How does a closer flame eliminates the hydrophobicity at the soil surface? Please add more information to this statement and discuss. Add (possible) temperature ranges from the literature to prove your point.

L390 – How does the infiltration rate influences the amount of rainfall? The infiltration rate depends on the amount of rainfall (and soil properties). Please rewrite.

L393 – What are “emergency treatments or measures”? Which treatments or measures do the authors mean exactly?

L394 – The authors wrote that there no rain event occurred between the forest fire (April 4, 2019) and the sampling campaign (two months later). Do the authors have information about the first rainfall event after their sampling campaign? Did they observe any rainfall erosion? Please be more accurate.

Please do not mention the tables and figures again in the Discussion part (except if they present synthesized information).

5. Conclusions

L398 – I suggest writing “after a high-intensity forest fire” instead of “after the fire”.

L400 – Please write “P. densiflora” instead of “Pinus densiflora”.

L412 – What are “emergency treatments”? (see comment on L393)

L402-411 – This is mainly a summary and not a conclusion of the main points from the Discussion part. Please rewrite.

References

The publication year of some references are shown in bold, other not. Why?

Some references are shown without any information about the pages and publishing company (e.g., Ref. 71, 78, 80).

L436 – Please add a space between “in” and “Pinus”.

L648 – Please check if the colon after “Sons” is correct.

Round 2

Reviewer 2 Report

None

Author Response

Dear Reviewer,

We would like to thank you for reviewing. With your comments, we checked the manuscript and revised typos, and rephrased when needed.

And, we also rephrased the sections to reflect the reviewer comments, especially Abstract, Discussion, and Conclusion.

Reviewer 3 Report

A report for the revised manuscript forests-1181037, entitled “Post-fire Impacts of Vegetation Burning on Soil Properties and Water Repellency in a Pine Forest, South Korea”

****General comments****

The authors have improved their manuscript. It is very much appreciated that they have made an effort to respond to the reviewer’s comments and questions.

However, the authors did not respond to every comment, especially the general comments about the abstract, introduction and conclusions. Unfortunately, the conclusions are still a summary, and do not provide a synthetic view about the discussed results. The authors fail to place their results into the bigger picture. Why is it important to know if forest fires alter the soil properties and soil water repellency? How would a permanent soil hydrophobicity affect the ecosystem dynamics? How long lasts a soil hydrophobicity? The authors present the results but do not connect them to the forest ecosystem. Many questions remain unanswered to the reader after reading the discussion and conclusions.

The discussion still does not contain EC or bulk density, which have been mentioned in the Results section.

I pronounce the importance of checking the (revised) manuscript very thoroughly for editorial and content (especially repetitions) errors before submission.

Hence, the manuscript needs to be revised again.

****Specific comments****

Title

---

Abstract

Please add the forest type (“Pinus densiflora”), the soil depth sampled, the analyzed soil characteristics (TOC, pH, EC, bulk density, CEC), number of soil samples, slope gradient types (steep, mild, gentle), and a final perspective/conclusion/outlook

Keywords

---

1. Introduction

Why is it important to study the soil hydrophobicity? What is the possible danger/benefit for a forest ecosystem? Any consequences for the soil and forest ecosystem or perhaps for the hydrological cycle? Which forest soil dynamics could be affected by a fire-induced soil hydrophobicity? How long does a water repellent layer last?

Please add information why TOC, pH, bulk density, etc. will be analyzed as well (state-of-the-art information, relation to forest (soil) ecosystem). Those soil characteristics/variables are not mentioned in the introduction.

L35-36 – Please write “can cause irreparable economic and ecological impacts on …”.

L35 – Please be aware that fire does not always cause a “irreparable” loss to the forest ecosystem.

L39 – Please write “conditions” (plural).

L39 – Please delete “seasons”. It is enough to write “winter and spring”.

L40-41 – Please add “South” to the sentence.

L49-50 – This paragraph begins with the same argument as the first paragraph (L35).

L66 – I suggest combining the paragraph L60-65 with the paragraph beginning at L66.

L98 – The term “captured the world’s attention” should be reconsidered as it sounds more like a comic term than a scientific one. I suggest beginning the paragraph with the sentence from L100.

L102 – Why do the authors use the term “non-Mediterranean forests”? Why do they make such a distinction?

L104 – Do the authors mean “North America” or both continents (North and South America)?

L105 – Please add the few studies that dealt with soil water repellency in Asian monsoon climate regions.

L106-109 – This sentence is almost identical to the next sentence about the objectives. Please rewrite.

L113 – Please write “on the east coast” instead of “in the east coast”.

2. Materials and Methods

L116-117 – Figure 1a should be mentioned here.

L129 – What is a “principle soil order”? Do the authors mean “dominating soil type”?

L130 – What do the authors mean by “bedrock underlies sedimentary rock”? Please add the type of parent material (from which the Inceptisol developed). Please add e.g. “US soil taxonomy” or “WRB” to the soil type.

L133 – Please delete the comma after “Q.” when the authors write “Quercus”.

L135 – Please write “have” instead of “has”.

L137 – Please write “thickness” instead of “thicknesses”.

L139 – I suggest showing or mentioning “Gangwon Province” in Figurea 1a.

L139 – Please delete “were” and write e.g., “occurred”.

L141 – Sokcho and Donghae are cities? If so, please delete the word “cities” from the sentence.

Figure 1 – I suggest rewriting the figure captions. Suggestion: “(a) Location of the three large fires on April 4, 20219 (Goseng, Inje, Okgye); (b) location of 15 (?) study sites in the study area Okgye including the fire severity levels (crown-fuel consumption, foliage necrosis, surface-fuel consumption) (NIFoS South Korea and Won et al. [51]).

Figure 1a – Please write “Okgye (study area)” instead of “study site”. The sites are the ones in blue shown in Figure 1b.

Figure 1a- Please add the word “Gangwon Province” to the map or the figure captions.

Figure 1a and 1b - Please find a more distinct symbol for the locations. It is difficult to detect the small symbols, especially when there are two (or more) at almost the same spot.

Figure 1b – I counted 15 study sites. Is this correct?

L151 – Please write “three days after the fire” instead of “after three days of the fire”.

L162 – It was only one fire event on April 4, 2019 in the Okgye forest, correct?! Please write “fire event” (singular) instead of “fire events”.

L165 – Please delete “were”.

L178-183 – Figure 1b could be mentioned in this paragraph.

L205-206 – Please add the diameter and length of the soil sampling devise.

L207-208 – Is this sentence one single paragraph? Why?

L213 – Please write “representation” instead of “represent”.

L217 – Please add information why the author decided to not use 5-10 cm, and 15-20 cm. The DIK-1601 is a soil corer; hence it samples a complete soil core from 0 to 20 cm.

Add information on what happened to the soil samples in the lab. Were they dried (temperature) or sieved (mesh size)? Add information on the analyzing device for the TOC content.  

L265 – I suggest writing “to determine differences” or “to detect differences” instead of “to test differences”.

3. Results

L275 – Which soil taxonomy was used? Please add information.

L276-280 – I suggest to round the numbers to one digit after the comma.

L280 - Please add “respectively” after the numbers.

L284 – Please write “p” in italics.

L289 – Please replace “bulk density for” with “bulk density of”.

L293 – Please add “average” or “mean” to indicate that these are the mean values. The authors could also add that they do not show the SE or SD due to the graphic type.

L294 – Please delete “was”.

L297 – Please delete “area”.

L298 – Please delete “that” and “areas”.

L299-300 – Please delete “areas”.

L300 – Please rewrite “were not”.

Figure 4 – I suggest using different line types (dotted, slash, etc.) and different symbols (circle, box, star, etc.).

L305 – Please write “in FN and CC soils compared to unburned soils”.

L306-307 – That is an interpretation and does not belong to the Results section.

L308 – Please delete “of soils”.

L309 – Please write “at 10-15 cm soil depth” instead of “at the 10-15 cm”.

L313 – Can the authors express “the greatest difference” in numbers, please?

L320-322 – This sounds like an interpretation of the results/explanation for the results. Please delete. In addition, the authors do not know the pre-fire organic matter content.

L332 – Do the authors mean “spread” as vertically/spatially spread?

L341 – Please write “and was excluded from the figure” instead of “and excluded in the figure”.

L352 – Please replace “revealed” with “showed”.

L370 – Please add “the” between “than” and “other”.

L380 – Please add “the” between “have” and “largest”.

L386-387 – This has been mentioned already in L378.

L388-389 – This has been mentioned already in L376-377.

4. Discussion

L394-395 – This paragraph will be combined with the next paragraph (L402-410)?

L404-405 – Please be careful with the tenses. Please use either present tense (“seems”) or past tense (“agreed”). Choose a uniform style.

L413-414 – Please be accurate when writing “compounds”. Which compounds do the authors mean?

L411-414 – That is a very short discussion about TOC. The authors hardly explain the decrease of TOC with soil depth, but focus on the quality of TOC which they haven’t analyzed. Please add more information and discuss the role of TOC in a fire-affected forest ecosystem.

L416 – Please add “the” between “to” and “decrease”.

L417 – Please be more accurate about “lower soil heating”. Lower compared to which temperature? How do pyrogenic organic compounds affect the soil pH? The authors fail to discuss the pH of SC, which is much higher than for UB, FN and CC.

L418-420 – Please be more accurate about the relationship between soil pH and oxides, hydroxides and organic carbon denaturation.

L419 – Please rewrite this sentences (especially “increase as the result of oxides,…”, which makes no sense).

L422-423 – Please be more specific about the relationship between CEC and clay minerals.

L423 – Please explain how CEC depends on the temperature.

The authors fail to discuss the effect of fire on EC and bulk density.

What is the final statement of the discussion of fire on soil properties? Please add this to this chapter.

L433-434 – Please explain the “circumstances”.

L436-442 – This has been discussed (in parts) in Chapter 4.1 already.

L444 – Please delete “to”.

L447 – Please write “condense” instead of “condensed”. The authors haven’t measure this, hence it is a suggestion based on other published studies.

L457-464 – Please add information about the temperature threshold instead of writing “higher temperature”.

L470-473 – These two sentences basically contain the same information from L468-470.

L477 – Please add “time” after “residence”.

L478 – This has already been mentioned in L 474.

5. Conclusions

The authors apparently haven’t considered the general comments in which the low quality of the Conclusions was mentioned. The presented Conclusions is not a conclusion, but a summary. Please rewrite completely. A conclusion always contains the most important outcomes of the study and what do the outcomes mean for the forest ecosystem or future studies.

References

---

Author Response

Dear Reviewer,

We would like to thank your efforts for careful and thorough reading of this manuscript and thoughtful comments and constructive suggestions, which help to improve the quality of this manuscript. We have been able to incorporate changes to reflect most of your comments. We have highlighted the changes within the manuscript by using “Track Changes”. Here is a point-by-point response to the reviewers’ comments and concerns. Please see the attachment.
